# E-cigarette use among university students in Palestine: Prevalence, knowledge, and determinant factors

**Mustafa Ghanim**[1]*, **Maha Rabayaa**[1], **Mohammad Abuawad**[1], **Munther Saeedi**[2], **Johnny Amer**[3]

**1** Faculty of Medicine and Health Sciences, Department of Biomedical Sciences, An-Najah National University, Nablus, Palestine, **2** Language Centre, An-Najah National University, Nablus, Palestine, **3** Faculty of Medicine and Health Sciences, Department of Allied and Applied Medical Sciences, An-Najah National University, Nablus, Palestine

* mustafa.ghanim@najah.edu

**Data Availability Statement:** All relevant data are within the manuscript and its Supporting Information files.

## Abstract

### Background

Recent reports indicated accelerated rates of e-cigarette use, especially among youth in various Middle Eastern countries, including Palestine. Nevertheless, little is known about knowledge, attitudes, and perceptions regarding this topic in Palestine. This study aimed to assess the prevalence of e-cigarette use among Palestinian university students, along with their knowledge, attitudes, and perceptions about e-cigarette use.

### Methods

An observational cross-sectional study, utilizing an online self-administered questionnaire, was conducted on Palestinian students from five universities between 17/04/2023 and 04/11/2023.

### Results

A total of 1002 Palestinian university students completed the questionnaire. The prevalence of e-cigarette use among students was 18.1%. The mean knowledge score about e-cigarettes was significantly lower among the users of e-cigarettes compared to non-users. E-cigarette use was significantly associated with the participants' smoking status. Among e-cigarette users, 43.6% were also current traditional cigarette users, and 66.9% were current waterpipe users. E-cigarette use was significantly associated with having a friend who is a smoker and/ or a smoking mother. Binary logistic regression revealed a significant positive effect between the participant's smoking status, the mother's smoking status, knowledge about e-cigarettes, and the use of e-cigarettes (p-value < 0.05). Among e-cigarette users, 18.8% used them in indoor places at the university, and 25% reported using them daily in the past month. Affordability of e-cigarettes was the most reported reason for their use (47.5%).

**Funding:** The author(s) received no specific funding for this work.

**Competing interests:** The authors have declared that no competing interests exist.

## Conclusion

This study concluded that e-cigarette use is prevalent and rapidly rising among university students in Palestine. This is worrisome as it is significantly associated with insufficient knowledge about the adverse health effects of E-cigarette use, and its addictive nature. These findings focus on the importance of improving the students' knowledge about e-cigarette use by implementing educational campaigns and considering age regulations on e-cigarette availability and use.

## 1. Introduction

Electronic cigarettes (e-cigarettes) are devices that operate by heating a solution containing humectants, nicotine (in most cases), and flavorings, that create an aerosol. Inhalation of this aerosol is referred to as electronic smoking (e-smoking) [1,2].

E-cigarettes are becoming increasingly popular globally, especially among young adults [3]. According to the National Center for Chronic Disease Prevention and Health Promotion, e-cigarette use was reported as the second most commonly used nicotine product in the United States in 2020–2021 [4]. In the United States, 4.5% of adults reported current e-cigarette use in 2021 [4]. Even more concerning, around 10% of middle and high school students were e-smoking in 2023 [5]. E-cigarette use prevalence in Europe varies widely, ranging from 0.2% to 27% [6]. In the Arab world, a study conducted among students from three universities in the United Arab Emirates revealed that 23% of participants used e-cigarettes [7]. Another study from Saudi Arabia reported that 27.7% of health sciences university students were e-cigarette users [8]. Among university students in Qatar, it was reported that 14% of students were e-cigarette users [9]. In Jordan, recent studies indicated that the prevalence of e-cigarette use among university students ranged between 11.7% and 18% [10,11].

Several factors contribute to this rapid rise in e-cigarette use among young adults. There is a widespread misconception that e-cigarette use is less harmful than traditional smoking [12,13]. E-cigarette use is marketed as an effective alternative practice to reduce the conventional cigarette consumption [9,14]. Marketing messages claim that e-cigarette use is safer and cleaner than traditional cigarettes [15,16]. E-cigarette availability, the possibility of using them in more places, peer influence, and curiosity are among additional factors that encourage e-cigarette use [17]. Some individuals consider e-cigarette use to be effective for smoking cessation [15,18–20].

Recent reports have shown that e-cigarette use is not devoid of risk [21]. There is evidence of a similar or higher addictive potential of e-cigarettes compared to traditional smoking [8,22]. E-cigarette use induces airway inflammation, lung injury, and ciliary dysfunction, and increases mucus secretion [23,24]. E-cigarette users are at a higher risk of developing acute and chronic health conditions, including stroke, myocardial infarction, coronary artery diseases, and atherosclerosis when compared to non-users [25–27].

The high prevalence of traditional smoking among Palestinian university students was extensively reported, which varied from 22.8% to 56% [28–31]. Nevertheless, limited research has been conducted on e-cigarette use. Two samples from a single university demonstrated that the prevalence of e-cigarette use among students was 4.6% and 13.3% [32,33].

Recently, Nazzal et al., reported that the prevalence of e-cigarette use was high (19.7%) among students recruited from six Palestinian universities [34]. Moreover, Nazzal found that students' knowledge about e-cigarette use was suboptimal, with misconceptions

regarding their safety and health effects, in addition to the presence of negative attitudes towards e-cigarette use. This current study aims to assess the prevalence, knowledge, attitudes, and perceptions of harm related to e-cigarette use among Palestinian students recruited from five large universities.

## 2. Methods

### 2.1 Study design and sampling

This was a cross-sectional study conducted utilizing a web-based survey between 17/04/2023 and 04/11/2023 amongst Palestinian university students. Given the overall count of Palestinian university students of approximately 120 thousand students all over the area of the West Bank in which there are around 3.2 million inhabitants, the sample size was determined by a 95% confidence interval (CI) and an accepted margin of error of 5%. The minimum sample size was determined as 383 participants using an online sample size calculator (www.raosoft.com). A convenient sample of 1002 students was enrolled in the study following their agreement to participate. The study included Palestinian university students, whether they were e-cigarette users or not. Students who had not agreed to participate and who had not completed the questionnaire were not included in the study.

### 2.2 Ethical consideration

The study was approved by the Institutional Review Board (IRB) office at An-Najah National University (Ref. Med. April 2023/3). Informed consent was obtained from all participants before they could proceed to the online questionnaire questions. The consent form outlined the study's purpose and ensured that participation was optional and anonymous, and there were no repercussions for non-participation.

### 2.3 Study tools

Employing a pre-tested questionnaire in Arabic from prior research conducted among university students in Qatar, with clear consent obtained from the author [35]. The questionnaire was initially adapted from the Global Adult Tobacco Survey and the American Cancer Society's Tobacco-Free Generation Campus Initiative: Cohort 5 Student Survey (2020–2021) [36]. The Arabic version was used in the current study and the calculated Cronbach's alpha value for internal consistency was 0.74. Furthermore, the Arabic version of the questionnaire underwent evaluation by five language experts and it was found to be consistent and achieve the target of the study. Additionally, the Kaiser-Meyer-Olkin Measure of Sampling Adequacy indicated that the sample size of the study was appropriate for the factor analysis (0.74), which should be greater than 0.60 [37] to conduct a factor analysis. Bartlett's Test of Sphericity was also significant (Sig. = 0.000<0.05) which means that our variables are related, thus deemed suitable for structure detection and for conducting a factor analysis. The cumulative variance explained by two factors is 53.123% of the total variance as shown from the table below.

  The questionnaire was constructed using Google Forms and distributed online using students' sites courses and universities' official e-learning websites. It consisted of four sections. The first section collected socio-demographic information, including age, gender, marital status, specialty, study year, family income, and place of residence. In the second section, participants were asked about their current smoking status, as well as that of their family and friends, and assessed the prevalence of various nicotine products among university students. The third part assessed students' understanding regarding the health effects of e-cigarettes, encompassing seven statements: "E-cigarettes can cause lung cancer," "E-cigarettes can cause

| Total Variance Explained | | | | | | |
|---|---|---|---|---|---|---|
| Knowledge item | Initial Eigenvalues | | | Rotation Sums of Squared Loadings | | |
| | Total | % of Variance | Cumulative % | Total | % of Variance | Cumulative % |
| 1 | 2.327 | 33.250 | 33.250 | 2.283 | 32.620 | 32.620 |
| 2 | 1.391 | 19.876 | 53.126 | 1.435 | 20.505 | 53.126 |
| 3 | 0.910 | 13.001 | 66.127 | | | |
| 4 | 0.671 | 9.592 | 75.719 | | | |
| 5 | 0.642 | 9.171 | 84.890 | | | |
| 6 | 0.583 | 8.325 | 93.215 | | | |
| 7 | 0.475 | 6.785 | 100.000 | | | |

Extraction Method: Principal Component Analysis.

cardiovascular problems," "E-cigarettes can cause cerebral stroke," "E-cigarettes do not contain carcinogenic ingredients," "E-cigarettes are addictive," "E-cigarettes are less harmful to health compared to traditional cigarettes," and "E-cigarettes prevent one from smoking traditional cigarettes." The total knowledge score was computed by summing the correct answers, assigning a value of one to each correct response and zero to "incorrect" or "don't know" answers. Consequently, the knowledge score ranged from 0 to 7 based on the number of correct answers given by the participant. The fourth section specifically targeted e-cigarette users, exploring their practices and reasons for using e-cigarettes. The participant can select more than one reason for using e-cigarettes.

## 2.4 Statistical analysis

The statistical analyses were conducted using Statistical Package for the Social Sciences version 21 (SPSS 21) by IBM Corp., Armonk, N.Y., USA. Descriptive analyses were utilized for all variables, employing mean and standard deviation for age and e-cigarette use knowledge score. Bivariate analyses were employed to examine the associations between e-cigarette use and each knowledge item, knowledge total score, and the current smoking status of family members or friends. For categorical variables, the Chi-square test was employed, while the t-test was used for scale variables (knowledge score). Variables showing significance at the bivariate level were incorporated in a binary logistic regression model to identify the potential determinants of e-cigarette use.

## 3. Results

### 3.1 Sample characteristics and nicotine products use

A total of 1002 Palestinian university students participated in the study, with a mean age of 21.06 years. The majority were single (76.8%), female (58.6%), undergraduate students (87.6%), residents of villages (48.4%), pursuing medicine degrees (28.8%), and in their first year of study (26.9%). More than 40% reported having a father who smokes, while only 9% reported having a mother who smokes. Over a third reported having at least one smoking sibling, and a similar proportion reported having a smoking friend. Concerning their smoking status, 32.1% identified as smokers of any nicotine product form, with 16% using traditional cigarettes, 27.6% using water pipes, and 18.1% using e-cigarettes. The descriptive statistics are detailed in Table 1.

**Table 1. Demographic and basic characteristics of the participants.**

| Variable | | n | % |
|---|---|---|---|
| Age | Mean ±SD | 21.06±3.41 | |
| Age groups | 20 or less | 559 | 55.8 |
| | 21–23 | 311 | 31 |
| | 24 or more | 132 | 13.2 |
| Marital status | Single | 770 | 76.8 |
| | Married | 121 | 12.1 |
| | Others | 111 | 11.1 |
| Educational level | Undergraduate | 878 | 87.6 |
| | Master | 64 | 6.4 |
| | PhD | 60 | 6 |
| Gender | Male | 415 | 41.4 |
| | Female | 587 | 58.6 |
| Place of residence | City | 416 | 41.5 |
| | Village | 485 | 48.4 |
| | Camp | 101 | 10.1 |
| Monthly family income (dollars) | Less than 516 | 101 | 10.1 |
| | 516–1033 | 429 | 42.8 |
| | More than 1033 | 472 | 47.1 |
| Specialty/Collage | Science and art | 166 | 16.6 |
| | Health Sciences | 100 | 10 |
| | Medicine | 289 | 28.8 |
| | Pharmacy | 21 | 2.1 |
| | Dentistry | 35 | 3.5 |
| | Islamic studies | 58 | 5.8 |
| | Business and economics | 90 | 9 |
| | Education | 101 | 10.1 |
| | Engineering and IT | 73 | 7.3 |
| | Law | 69 | 6.8 |
| Year of Study | First | 270 | 26.9 |
| | Second | 215 | 21.5 |
| | Third | 199 | 19.9 |
| | Fourth | 207 | 20.7 |
| | Fifth | 57 | 5.7 |
| | Sixth | 54 | 5.4 |
| **Current smoking status of family and friends** | | | |
| My father is a current smoker | Yes | 421 | 42 |
| | No | 581 | 58 |
| My mother is a current smoker | Yes | 90 | 9 |
| | No | 912 | 91 |
| At least one of my siblings is a smoker | Yes | 358 | 35.7 |
| | No | 644 | 64.3 |
| At least one of my friends is a smoker | Yes | 377 | 37.6 |
| | No | 625 | 62.4 |
| **Current smoking status of the participants** | | | |
| Are you smoker | Yes | 322 | 32.1 |
| | No | 680 | 67.9 |
| Do you use a traditional cigarette? | Yes | 160 | 16 |
| | No | 842 | 84 |
| Do you use water pipes | Yes | 277 | 27.6 |
| | No | 725 | 72.4 |
| Do you use e-cigarettes | Yes | 181 | 18.1 |
| | No | 821 | 81.9 |

## 3.2 Knowledge about the health risks of e-cigarettes among all participants and the association between knowledge and e-cigarette use

The majority of participants concurred that e-cigarettes can cause lung cancer, cardiovascular problems, and cerebral strokes (80.1%, 76.9%, and 61.3%, respectively). About 72.7% agreed that e-cigarettes are addictive. However, 73% disagreed with the notion that e-cigarettes lack carcinogenic ingredients. When it comes to the perception that e-cigarettes are less harmful than traditional cigarettes, only 46.2% disagreed, while 27.6% and 26.1% answered 'agree' and 'I do not know', respectively. Regarding the belief that e-cigarettes can prevent one from smoking traditional cigarettes, 36.8% disagreed, whereas 32.7% and 30.4% responded 'agree' and 'I do not know,' respectively. The mean knowledge score for correct answers was 4.47.

There was a statistically significant association between e-cigarette use and all knowledge items and the total knowledge score about e-cigarettes. In comparison to non-users of e-cigarettes, fewer e-cigarette users agreed that e-cigarettes can cause lung cancer (86.0% and 53.6%, respectively), cardiovascular disease (84.4% and 43.1%, respectively), or cerebral stroke (68.6% and 28.2%, respectively). Additionally, 80.8% of non-e-cigarette users agreed that e-cigarettes were addictive, while only 35.9% of e-cigarette users agreed. A lower percentage of e-cigarette users believed that e-cigarettes do not contain carcinogenic ingredients (78.6% and 47.5%, respectively) and that e-cigarettes were less harmful than traditional cigarettes (49.7% and 30.4%, respectively), in comparison to non-e-cigarette users. Moreover, 38.5% of non-e-cigarette users disagreed that e-cigarettes could help prevent smoking traditional cigarettes, while 29.3% of e-cigarette users disagreed. The mean total knowledge score about e-cigarettes was significantly lower among e-cigarette users (2.68) compared to non-users (4.86). The detailed results are presented in Table 2.

## 3.3 E-cigarette users' practices and reasons for e-cigarette use

The mean age of initiation for e-cigarette use was 20.38 years. Approximately 25% of e-cigarette users reported using them daily in the past month. About 32.6% do not use e-cigarettes

**Table 2. Knowledge about e-cigarettes among all respondents and the association between knowledge and e-cigarette use.**

| Knowledge item | All participants (n = 1002) n (%) | | | E-cigarette users (n = 181) n (%) | | | Non-e-cigarettes users (n = 821) n (%) | | | |
|---|---|---|---|---|---|---|---|---|---|---|
| | agree | disagree | IDK | agree | disagree | IDK | agree | Disagree | IDK | p-value |
| 1. E-cigarettes can cause lung cancer. | 803 (80.1) | 85 (8.5) | 114 (11.4) | 97 (53.6) | 44 (24.3) | 40 (22.1) | 706 (86) | 41 (5) | 74 (9) | <**0.001** |
| 2. E-cigarettes can cause cardiovascular problems. | 771 (76.9) | 86 (8.6) | 145 (14.5) | 78 (43.1) | 47 (26) | 56 (30.9) | 693 (84.4) | 39 (4.8) | 89 (10.8) | <**0.001** |
| 3. E-cigarettes can cause cerebral stroke. | 614 (61.3) | 105 (10.5) | 283 (28.2) | 51 (28.2) | 52 (28.7) | 78 (43.1) | 563 (68.6) | 53 (6.5) | 205 (25) | <**0.001** |
| 4. **E-cigarettes do not contain carcinogenic ingredients.** | 83 (8.3) | 731 (73) | 188 (18.8) | 34 (18.8) | 86 (47.5) | 61 (33.7) | 49 (6) | 645 (78.6) | 127 (15.5) | <**0.001** |
| 5. E-cigarettes are addictive. | 728 (72.7) | 122 (12.2) | 152 (15.2) | 65 (35.9) | 50 (27.6) | 66 (36.5) | 663 (80.8) | 72 (8.8) | 86 (10.5) | <**0.001** |
| 6. **E-cigarettes are less harmful to health compared to traditional cigarettes.** | 277 (27.6) | 463 (46.2) | 262 (26.1) | 69 (38.1) | 55 (30.4) | 57 (31.5) | 208 (25.3) | 408 (49.7) | 205 (25) | <**0.001** |
| 7. **E-cigarettes prevent one from smoking traditional cigarettes.** | 328 (32.7) | 369 (36.8) | 305 (30.4) | 74 (40.9) | 53 (29.3) | 54 (29.8) | 254 (30.9) | 316 (38.5) | 251 (30.6) | **0.019** |
| Knowledge (mean± SD) | 4.47±1.99 | | | 2.68±2.04 | | | 4.86±1.75 | | | <**0.001** |

*P-value; Chi-square test for categorical variables and t-test for continuous variables. IDK: I do not know.* **The correct answers to the bolded questions are false.**

on campus, while only 18.8% use them indoors at the university. 32% reported using e-cigarettes in social situations, 18.2% used them in stressful situations, and 22.7% used them during university hours. Regarding reasons for using e-cigarettes among university students, affordability was the most cited reason (47.5%), 36.5% mentioned using e-cigarettes in places where traditional cigarettes are not allowed, 34.3% reported using e-cigarettes because they believed that e-cigarettes are less harmful than traditional cigarettes, 31.5% due to the availability of various e-cigarette flavors, 23.2% use e-cigarettes because they might be less harmful to people around them than traditional cigarettes. Less than 20% use them because e-cigarettes don't have a bad smell, might help them quit smoking, their friends use e-cigarettes, or simply appear cool. Results are shown in Table 3.

## 3.4 Association between participants' characteristics and smoking practices of family and friends and e-cigarette use

Table 4 indicates that the majority of e-cigarette users aged between 21 and 23 years old, were male, single, undergraduate, city residents, had a high family income (more than $1033 per month), studied at the College of Science and Art, and were in their third year of study. Significant variations in e-cigarette use were observed across all studied variables (p-value less than

**Table 3. E-cigarette users' practices and reasons for using the e-cigarettes (n = 181).**

| Variable | | n | % |
|---|---|---|---|
| age | Mean ±SD | 20.38±3.21 | |
| E-cigarettes used in the last 30 days | Daily | 46 | 25.4 |
| | twice a week | 44 | 24.3 |
| | Biweekly | 26 | 14.4 |
| | once per week | 29 | 16 |
| | once a month | 36 | 19.9 |
| Place of using e-cigarettes on campus | Indoors | 34 | 18.8 |
| | Outdoors | 54 | 29.8 |
| | both in and out | 34 | 18.8 |
| | Do not use E-smoking on campus | 59 | 32.6 |
| Timing of e-cigarette use | during university hours | 41 | 22.7 |
| | in social situation | 58 | 32 |
| | in stressful situation | 33 | 18.2 |
| | Others | 49 | 27.1 |
| **Reasons for using e-cigarettes** | | | |
| 1 | I use electronic cigarettes because they are affordable. | 86 | 47.5 |
| 2 | I use electronic cigarettes because I can use them in places where smoking cigarettes isn't allowed. | 66 | 36.5 |
| 3 | I use electronic cigarettes because they might be less harmful to me than smoking cigarettes. | 62 | 34.3 |
| 4 | I use electronic cigarettes because they might be less harmful to people around me than cigarettes. | 42 | 23.2 |
| 5 | I use electronic cigarettes because they come in flavors I like. | 57 | 31.5 |
| 6 | I use electronic cigarettes because they might help me quit smoking cigarettes. | 21 | 11.6 |
| 7 | I use electronic cigarettes because they do not smell. | 31 | 17.1 |
| 8 | I use electronic cigarettes because my friends use them. | 14 | 7.7 |
| 9 | I use electronic cigarettes because I look cool. | 11 | 6.1 |

**Table 4. Association between participants' characteristics and smoking practices of family and friends and e-cigarette use.**

| | E-cigarette users (n = 181), n (%) | Non-e-cigarette users (n = 821), n (%) | P-value |
|---|---|---|---|
| **Participants characteristics** | | | |
| Age group | | | |
| 20 or less | 55 (30.4) | 504 (61.4) | <0.001 |
| 21–23 | 76 (43) | 235 (28.6) | |
| 24 or more | 50 (27.6) | 82 (10) | |
| Gender | | | |
| Male | 107 (59.1) | 308 (37.5) | <0.001 |
| Female | 74 (40.9) | 513 (62.5) | |
| Marital status | | | |
| Single | 104 (57.5) | 666 (81.1) | <0.001 |
| Married | 49 (27.1) | 72 (8.8) | |
| Others | 28 (15.5) | 83 (10.1) | |
| Educational level | | | |
| Undergraduate | 130 (71.8) | 748 (91.1) | <0.001 |
| Master | 27 (14.9) | 37 (4.5) | |
| PhD | 24 (13.3) | 36 (4.4) | |
| Place of residence | | | |
| City | 75 (41.4) | 341 (41.5) | <0.001 |
| Village | 64 (35.4) | 421 (51.3) | |
| Camp | 42 (23.2) | 59 (7.2) | |
| Monthly family income (dollars) | | | |
| Less than 516 | 29 (16) | 72 (8.8) | 0.014 |
| 516–1033 | 72 (39.8) | 357 (43.5) | |
| More than 1033 | 80 (44.2) | 392 (47.7) | |
| Specialty/collage | | | |
| Science and art | 38 (21) | 128 (15.6) | <0.001 |
| Health Sciences | 10 (5.5) | 90 (11) | |
| Medicine | 11 (6.1) | 278 (33.9) | |
| Pharmacy | 0 (0) | 21 (2.6) | |
| Dentistry | 4 (2.2) | 31 (3.8) | |
| Islamic studies | 19 (10.5) | 39 (4.8) | |
| Business and economics | 21 (11.6) | 69 (8.4) | |
| Education | 26 (14.4) | 75 (9.1) | |
| Engineering and IT | 23 (12.7) | 50 (6.1) | |
| Law | 29 (16) | 40 (4.9) | |
| Year of Study | | | |
| First | 21 (11.6) | 249 (30.3) | <0.001 |
| Second | 41 (22.7) | 174 (21.2) | |
| Third | 42 (23.2) | 157 (19.1) | |
| Fourth | 36 (19.9) | 171 (20.8) | |
| Fifth | 17 (9.4) | 40 (4.9) | |
| Sixth | 24 (13.3) | 30 (3.7) | |
| **The smoking status of the participant, Family, and friends** | | | |
| You are using traditional cigarettes | 79 (43.6) | 81 (9.9) | <0.001 |
| You are using water pipes | 121 (66.9) | 156 (19) | <0.001 |
| Your father is a smoker | 81 (44.8) | 340 (41.4) | 0.229 |
| Your mother is a smoker | 39 (21.5) | 51 (6.2) | <0.001 |
| At least one of my siblings is a smoker | 70 (38.7) | 288 (35.1) | 0.203 |
| At least one of my friends is a smoker | 88 (48.6) | 289 (35.2) | 0.001 |

0.05). The smoking status of participants was significantly associated with e-cigarette use. Among e-cigarette users, 43.6% were also traditional cigarette users, and 66.9% were waterpipe users. E-cigarette use was significantly associated with having a friend who is a smoker and a smoking mother. However, there was no significant association between e-cigarette use and the smoking status of fathers or siblings. The results are presented in Table 5.

### 3.5 Determinants of e-cigarette use

A binary logistic regression was conducted to identify the potential determinants of e-cigarette use. The variables related to the smoking status of the participants, family, and friends, along with knowledge scores that showed significance at the bivariate analysis level, were included in the regression model. A positively significant effect was found between the participant's smoking status, the mother's smoking status, knowledge about e-cigarettes, and the use of e-cigarettes (p-value < 0.05). The odds ratios for being a non-smoker in general, a non-traditional cigarette user, and a non-waterpipes user were 3.22, 2.3, and 3.2, respectively, as determinants for being a non-e-cigarette user. Additionally, when the mother is a non-smoker, it is more likely that the participant is a non-e-cigarette user, with an odds ratio of 2.65. A higher knowledge about e-cigarettes is a positively significant determinant for being a non-e-cigarette user, with an odds ratio of 1.71. Results are shown in Table 5.

## 4. Discussion

The current research is a leading one among a few related previous studies that investigated the prevalence, knowledge, and attitudes toward e-cigarette use among university students from several universities in Palestine. Our results indicated that the prevalence of e-cigarette use among Palestinian university students is 18.1%. This prevalence is higher than what was reported in two prior Palestinian studies which were 13.3% [32]. On the other hand, our results are similar to what was reported from a recent study conducted on a larger sample of Palestinian students that was conducted on several universities in which the prevalence of e-cigarette use was 19.7% [34]. These discrepancies in prevalence could be attributed to the sample size, sampling methods, and sociodemographic differences. It is worth mentioning that these two later studies were conducted among students from a single university while our study included students from five different universities.

The prevalence of e-cigarette use in our study was higher than what was reported in the neighboring countries including Qatar (14%), and Jordan (11%) [9,38]. It is noteworthy that the female-to-male ratio in these studies was much higher than that in our study. This is consistent with what was reported previously that e-cigarette use is predominantly practiced by young males [39]. On the contrary, the prevalence of e-cigarette use in our study was lower than the reported prevalence among university students from Saudi Arabia (33.8%) [40]. Notably, this later study included a smaller sample with an obviously higher male-to-female ratio in

**Table 5. Binomial logistic regression analysis of the association between independent variables from the bivariate analysis and e-cigarette use.**

| Variable (reference) | B | p-value | OR | 95% confidence | |
|---|---|---|---|---|---|
| | | | | Lower | Upper |
| You are a smoker (yes) | 1.168 | <**0.001** | 3.22 | 1.914 | 5.402 |
| You use traditional cigarettes (yes) | 0.835 | **0.001** | 2.3 | 1.412 | 3.759 |
| You use water pipes (yes) | 1.164 | <**0.001** | 3.2 | 2.008 | 5.108 |
| My mother is a smoker (yes) | 0.974 | **0.001** | 2.65 | 1.469 | 4.775 |
| At least one of my friends is a smoker (yes) | 0.285 | 0.189 | 1.33 | 0.869 | 2.035 |
| Knowledge score | 0.536 | <**0.001** | 1.71 | 1.566 | 1.866 |

comparison to our study. Our results support the notion that e-cigarette use is growing and gaining more popularity among the university Palestinian students.

The majority of participants were aware of the link between e-cigarette use and various health risks. Nevertheless, substantial knowledge gaps remain in crucial areas. More than half agreed or were neutral on the misconception that e-cigarette use is less harmful than traditional smoking. Similarly, two-thirds of participants believed or were neutral on the misconception that e-cigarette use helps quit smoking. Notably, a quarter of all participants and over 60% of e-cigarette users disagreed or were unaware of their addictive nature. These findings align with previous studies on university students from Qatar [9], Saudi Arabia [40], and Jordan [38].

The mean total knowledge score about e-cigarette use was significantly lower among participants who use e-cigarettes compared to non-users. Furthermore, there were significant differences in all studied knowledge items between e-cigarette users and non-users. These differences were most significant in items related to the association of e-cigarette use with health risks such as lung cancer, cardiovascular diseases, and stroke. These findings are in harmony with previous reports from Qatar, Jordan, Lebanon, Iraq, Kuwait, Turkey, and Saudi Arabia [9,38,41]. These knowledge gaps can contribute to the rapid increase in the prevalence of e-cigarette use among Palestinian university students. Moreover, this situation might encourage e-cigarette users to continue and influence their peers to initiate e-cigarette use. Thus, it is of utmost importance for stakeholders at the university, in society, and within the government to coordinate their efforts to enhance awareness among university students regarding e-cigarette use and its associated adverse effects.

The current study identified that the most important reasons for e-cigarette use were the affordability of these electronic devices, the ability to use them in more places, and the perception that e-cigarette use is less harmful than other forms of smoking. These findings are in line with several previous regional and international research in Jordan [11,38], Qatar [9], the USA [42], and Malaysia [43]. Previous studies on university students found that 78.6% and 34.8% from Qatar and Nepal, respectively were driven by the perceived benefit of e-cigarette use as a potential aid in smoking cessation [9,44]. However, only 11.6% of e-cigarette users in the current study cited the aid in smoking cessation as the reason for e-cigarette use. The current study identified a higher prevalence of e-cigarette use among males compared to females. This is in line with the prevalence reported among Indonesian adolescents where about 29% of males and only 6.3% of females were e-cigarette users [45]. Another recent study conducted in Qatar revealed also a higher prevalence of e-cigarette use among male students compared to females [9]. Previous reports suggested that e-cigarettes are mostly used by middle-aged current smokers, particularly males, to help them quit smoking [46]. It was reported that the interest in adopting healthy behaviour among Palestinian university students is more significant among females than males [47].

The present study found that e-cigarette use among Palestinian university students remains primarily used by current smokers. Indeed the majority (75.1%) of e-cigarette users were also regular smokers of other forms of nicotine products. This outcome aligns with findings from other studies [45]. E-cigarette use showed significant correlations with having a smoking friend, and also with having a smoking mother. These findings are in line with a study conducted among young adults in Thailand, revealing that having a partner using e-cigarettes increased the likelihood of e-cigarette use by 3.239 times [48]. Similarly, a study involving university students in Qatar reported consistent results regarding factors associated with using e-cigarettes and water pipes [9,49]. However, it could be difficult to confirm that smoking by mothers and friends will influence e-cigarette use when the two behaviors are co-occurring [50].

The current study stands as a leading example of comprehensive research on e-cigarette use among Palestinian university students in terms of sample, scope, and outcomes. Students from

five large Palestinian universities were recruited for the study. And a wide range of factors were compared between e-cigarette users and non-users. Nevertheless, the study had some limitations. First, as the used questionnaire was self-reported, misreporting and recall bias should be considered. Second, the cross-sectional nature of the study makes it difficult to drive cause-and-effect relationships.

This study concluded that e-cigarette use is prevalent and rapidly rising among university students in Palestine. This is worrying as it is significantly associated with several misconceptions and insufficient knowledge about the adverse health effects of E-cigarette use, and its addictive nature. The negative influence of peers and family members was highlighted as a significant explanation for the spread of E-cigarette use. Notably, the availability of E-cigarettes and the flexibility of using them in many places including indoors inside the university were the most commonly cited causes by E-cigarette users. These findings urge decision-makers to work beyond the scope of simply intensifying efforts to improve the student's knowledge about E-cigarette use, in addition to establishing regulations on E-cigarette availability and policies regarding its use in certain places.

## Supporting information

**S1 Raw data. Document containing the raw data of the study.**
(XLSX)

## Acknowledgments

The authors would like to thank An-Najah National University (www.najah.edu) for the technical support provided to publish the present manuscript. We would like to express our gratitude to Dr. Waleed Salameh, an expert in Educational English from the Faculty of Graduate Studies at An-Najah National University, for his invaluable assistance with the English editing of the revised manuscript.

## Author Contributions

**Conceptualization:** Mustafa Ghanim, Maha Rabayaa.

**Data curation:** Mustafa Ghanim, Maha Rabayaa, Mohammad Abuawad, Munther Saeedi, Johnny Amer.

**Formal analysis:** Mustafa Ghanim, Maha Rabayaa.

**Investigation:** Mustafa Ghanim, Maha Rabayaa, Johnny Amer.

**Methodology:** Mustafa Ghanim, Maha Rabayaa.

**Project administration:** Mustafa Ghanim.

**Supervision:** Mustafa Ghanim.

**Validation:** Mustafa Ghanim.

**Visualization:** Mohammad Abuawad, Munther Saeedi, Johnny Amer.

**Writing – original draft:** Mustafa Ghanim, Maha Rabayaa, Mohammad Abuawad, Munther Saeedi, Johnny Amer.

**Writing – review & editing:** Mustafa Ghanim, Maha Rabayaa, Mohammad Abuawad, Munther Saeedi, Johnny Amer.

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
