## [Decision Letter · Decision Letter 0]

5 Mar 2024

PONE-D-24-02812E-cigarettes among University Students in Palestine: Prevalence, Knowledge, and Determinant FactorsPLOS ONE

Dear Dr. Ghanim,

Thank you for submitting your manuscript to PLOS ONE. After careful consideration, we feel that it has merit but does not fully meet PLOS ONE’s publication criteria as it currently stands. Therefore, we invite you to submit a revised version of the manuscript that addresses the points raised during the review process.

Dear authors, Thank you for submitting your valuable work to Plos ONE journal. The reviewers raised some issues need to addressed. 

We look forward to receiving your revised manuscript.

Kind regards,

Mohammed Nasser Alhajj, BDS, MClinDent, PhD

Academic Editor

PLOS ONE

2. Please amend either the abstract on the online submission form (via Edit Submission) or the abstract in the manuscript so that they are identical.

Reviewers' comments:

Reviewer's Responses to Questions

**Comments to the Author**

1. Is the manuscript technically sound, and do the data support the conclusions?

Reviewer #1: Yes

Reviewer #2: Yes

Reviewer #3: Partly

2. Has the statistical analysis been performed appropriately and rigorously? 

Reviewer #1: Yes

Reviewer #2: I Don't Know

Reviewer #3: Yes

3. Have the authors made all data underlying the findings in their manuscript fully available?

Reviewer #1: Yes

Reviewer #2: Yes

Reviewer #3: Yes

4. Is the manuscript presented in an intelligible fashion and written in standard English?

Reviewer #1: No

Reviewer #2: Yes

Reviewer #3: No

5. Review Comments to the Author

Reviewer #1: This is a very important research study, and the first or second one to be carried in Palestine. I strongly recommend accepting this manuscript for publication; however, with minor revisions. I have two issues with this manuscript:

1- Methods: the authors need to describe how did they collect the data. It is not clear how did they distribute the survey.

2- Discussion: suggest to add a recent study about waterpipe use among university students:

Al-Jayyousi, G.F., Kurdi, R., Islam, N., Alhussaini, N. Z., Awada, S., & Abdul Rahim, H. (2022). Factors Affecting Waterpipe Tobacco Smoking among University Students in Qatar. Journal of Substance Use and Misuse, 57 (3), p. 392-401. Retrieved from: Doi: 10.1080/10826084.2021.2012695. Epub 2021 Dec 16. PMID: 34913828.

3- The manuscript needs language editing.

Thanks.

Reviewer #2: Thank you for the opportunity to review this manuscript.

This manuscript is mostly well written but needs some major revision which I believe will help to improve the quality of the paper.

1. The authors need to be specific when referring to data collected from a different context. For example US data is quoted without any reference to the US.

2. The title set out to investigate e-cigarette use but, in the manuscript, there is a lot of information about a comparison between e-cigarette use and tobacco smoking. This is confusing as it does not tally with the objective of the paper.

3. The authors used the term ‘e-smoking’ which is not traditionally used in the field. If they decide to use it in this paper, it should be introduced early on in the paper

4. Lines 55 – 61 refer to global data but give examples of only countries from the middle east. The authors can be specific to refer to data from the middle east which is a similar setting to where the data were collected so would be in order.

5. Lines 81 – 84: The author must know that e-cigarettes though usually classified WITH tobacco products, they are not tobacco products but nicotine products and should be referred to as nicotine products

6. Line 88: similarly, the use of e-cigarette is not referred to as “smoking” but just “use”. This should be corrected throughout the work

7. Line 97: the authors say this is the first of its kind research study done in palestine but a simple google search produced at least 2 recent studies with the same sample demographic and same country (see: (1) Jaber ME, Nouri L, Hamed A, et al. The epidemiology of electronic cigarette smoking among university students in the West Bank: Practice, motivation, and dependence of a new emerging hazard. Population Medicine. 2023;5(October):27. doi:10.18332/popmed/174287.; (2) Nazzal Z, Maraqa B, Azizeh R, Darawsha B, AbuAlrub I, Hmeidat M, Al-Jabari F. Exploring the prevalence, knowledge, attitudes and influencing factors of e-cigarette use among university students in Palestine: a cross-sectional study. BMJ Open. 2024 Feb 17;14(2):e080881. doi: 10.1136/bmjopen-2023-080881. PMID: 38367977; PMCID: PMC10875484.)

8. Methods section: The authors should add information about the number of students which made the final sample, how they were selected and what were the inclusion and exclusion criteria as well as the data collection procedure)

9. There are too many tables in this manuscript. Tables should be consolidated, and the manuscript should only have between 3 to 4 Tables.

10. The length of the manuscript can be reduced by between 10 to 20% to make this work more concise but with relevant information.

11. Line 193: correct table heading (bout to about)

12. Line 195: the authors mentioned negative answers as if they were not worked on before a scaled was consDRomamuli$49tructed. Negatively worded questions ought to have been reversed scored before a scale is formed. Kindly ensure this was done to avoid errors in the computation of the scale.

Reviewer #3: 1. Unify the used term throughout the manuscript: e-cigarettes OR e-cigarette OR e-smoking

Introduction:

1. Lines 51 – 54: The reference 4 is about prevalence of e-cigarette among high school students, not reference 5!!!!

2. Lines 54 – 55: The reference 6 is about prevalence among European countries, not globally!!!!

Methodology:

1. Lines 107 – 108: the sample size calculation is misleading!!!!! It is not indicated whether the response rate and the number of estimates were considered or not, not to mention the “design effects”!!!!! Further, the total population and the population frame were not referred to.

2. Lines 112 – 113: How can an online questionnaire be signed?????

3. Line 122 – 123: Merely mentioning that the questionnaire is translated into Arabic with good Cronbach alpha is not enough to use it on an Arabic population. In order to do so, you have to conduct a study on the psychometric properties of the translated version following very strict steps ahead of using it. This is a major methodological pitfall that jeopardizes the whole study.

4. Regarding the seven items that gauge the knowledge, I am afraid that there is no sound evidence for the correct responses!!!!!

5. What the category “Other” in the “Marital status” means in an Arabic Culture????

6. Based on what you chose the values of the family income as 516$, 516-1033$ and more than 1033$?????

Results:

1. Indicate that the responses to “Reasons for using e-cigarettes” are multiple choices.

2. Table 5: Correct the percentage 991.1%!!!!

3. Why you included the knowledge as a determinant in the logistic regression??? having a lower knowledge might be a consequence of being e-cigarette users, either as a direct association, or indirectly through ignoring and indifference its bad effects.

Discussion:

1. The discussion is superficial. For example, the authors didn’t comment on why there were differences in knowledge between e-cigarette users and non-users!!!!!!!

2. Many of the arguments are not supported by references!!!!!

3. Protrude your study’s strengths, and mention its limitations.

4. The conclusion is not relevant to the study at all.

6. PLOS authors have the option to publish the peer review history of their article (what does this mean?). If published, this will include your full peer review and any attached files.

Reviewer #1: No

Reviewer #2: No

Reviewer #3: **Yes: **Esam Halboub

---

## [Author Response · Author response to Decision Letter 0]

25 Mar 2024

Dear editor, PLOS ONE journal 23-03-2024

I would like to thank you as well as to thank the respected reviewers for their time and important comments which we believe they greatly enhanced the quality of our manuscript. Kindly find below our responses to the reviewer comments. Also, we highlighted all changes in the revised manuscript in red color.

Kind regards,

Mustafa Ghanim

Corresponding author on behalf of all authors

Reviewer #1: This is a very important research study, and the first or second one to be carried in Palestine. I strongly recommend accepting this manuscript for publication; however, with minor revisions. I have two issues with this manuscript:

1- Methods: the authors need to describe how did they collect the data. It is not clear how did they distribute the survey.

Response: The data collection method is added to the study tool section: ‘’ The questionnaire was constructed using Google Forms and distributed online using student's sites of courses and universities’ official e-learning websites.’’

2- Discussion: suggest to add a recent study about waterpipe use among university students:

Al-Jayyousi, G.F., Kurdi, R., Islam, N., Alhussaini, N. Z., Awada, S., & Abdul Rahim, H. (2022). Factors Affecting Waterpipe Tobacco Smoking among University Students in Qatar. Journal of Substance Use and Misuse, 57 (3), p. 392-401. Retrieved from: Doi: 10.1080/10826084.2021.2012695. Epub 2021 Dec 16. PMID: 34913828.

Response: Thank you for your suggestion. The reference is included in the discussion section as similar factors significantly associated with e-smoking are also associated with waterpipes used in the suggested reference. 

3- The manuscript needs language editing.

Response: Language editing was performed by a language expert.

Reviewer #2: Thank you for the opportunity to review this manuscript.

This manuscript is mostly well written but needs some major revision which I believe will help to improve the quality of the paper.

1. The authors need to be specific when referring to data collected from a different context. For example, US data is quoted without any reference to the US.

Response: The manuscript is revised and proper citations were added.

2. The title set out to investigate e-cigarette use but, in the manuscript, there is a lot of information about a comparison between e-cigarette use and tobacco smoking. This is confusing as it does not tally with the objective of the paper.

Response: The study aims to investigate the knowledge about e-cigarettes and the prevalence of e-cigarette use among Palestinian university students and the smoking practices of e-cigarette users. The comparison in the results was done between e-cigarette users and non-users to find the factors associated with e-smoking. Proper modifications were done.

3. The authors used the term ‘e-smoking’ which is not traditionally used in the field. If they decide to use it in this paper, it should be introduced early on in the paper

Response: The ‘e-smoking’ term is defined in the first paragraph of the introduction section as follows: ‘’The inhalation of this aerosol is referred to as electronic smoking (e-smoking)’’

4. Lines 55 – 61 refer to global data but give examples of only countries from the Middle East. The authors can be specific in referring to data from the Middle East which is a similar setting to where the data were collected so would be in order.

Response: The prevalence of e-smoking in different areas including the United States, Europe, and Arab countries in the introduction section to illustrate that e-smoking is a global concern. Your comment regarding the global prevalence is modified because the attached reference refers to the European population as follows: ‘’ E-cigarette use prevalence in Europe varies widely, ranging from 0.2% to 27% (6).’’

5. Lines 81 – 84: The author must know that e-cigarettes though usually classified WITH tobacco products, they are not tobacco products but nicotine products and should be referred to as nicotine products

Response: Nicotine product term is used instead of tobacco products all over the manuscript.

6. Line 88: similarly, the use of e-cigarette is not referred to as “smoking” but just “use”. This should be corrected throughout the work

Response: Modification is done. E-cigarette use and e-cigarette users’ terms were used instead of smoking terms throughout the manuscript.

7. Line 97: the authors say this is the first of its kind research study done in palestine but a simple google search produced at least 2 recent studies with the same sample demographic and same country (see: (1) Jaber ME, Nouri L, Hamed A, et al. The epidemiology of electronic cigarette smoking among university students in the West Bank: Practice, motivation, and dependence of a new emerging hazard. Population Medicine. 2023;5(October):27. doi:10.18332/popmed/174287.; (2) Nazzal Z, Maraqa B, Azizeh R, Darawsha B, AbuAlrub I, Hmeidat M, Al-Jabari F. Exploring the prevalence, knowledge, attitudes and influencing factors of e-cigarette use among university students in Palestine: a cross-sectional study. BMJ Open. 2024 Feb 17;14(2):e080881. doi: 10.1136/bmjopen-2023-080881. PMID: 38367977; PMCID: PMC10875484.)

Response: The first published study was just carried out on male students recruited from one single university and it was used as a reference in our study, however, our study involved both genders using a larger sample size from five different universities, and knowledge about e-cigarettes was not evaluated in the abovementioned study. 

Regarding the second-mentioned study, it was newly published in 2024 and our study was submitted in 2023 before it was available online. Additionally, the sentence was modified to reveal that our study is one of the leading studies concerning e-cigarettes in Palestine since it is not the first based on the mentioned recently published reference. Moreover, we have used a previously developed, translated and validated questionnaire while the second-mentioned study used a questionnaire developed by the authors. 

8. Methods section: The authors should add information about the number of students who made the final sample, how they were selected, and what the inclusion and exclusion criteria as well as the data collection procedure)

Response: The required details were added to the study design and sampling subsection of the methods. 

9. There are too many tables in this manuscript. Tables should be consolidated, and the manuscript should only have between 3 to 4 Tables.

Response: Tables 2 and 3 were merged in one table (Table 2) with proper modification in the results.

10. The length of the manuscript can be reduced by between 10 to 20% to make this work more concise but with relevant information.

Response: Proper modifications were done.

11. Line 193: correct table heading (bout to about)

Response: Correction is done

12. Line 195: the authors mentioned negative answers as if they were not worked on before a scale was consDRomamuli$49tructed. Negatively worded questions ought to have been reversed scored before a scale is formed. Kindly ensure this was done to avoid errors in the computation of the scale.

Response: The correct answers for 4,6, and 7 questions in the knowledge scale were intentionally false. Based on the questionnaire source instruction, every correct answer deserves one point while incorrect answers and ‘’I do not know’’ do not deserve any point. Negatively worded questions were reversed in scoring based on the scoring directions.

Reviewer #3: 1. Unify the used term throughout the manuscript: e-cigarettes OR e-cigarette OR e-smoking

Response: The ‘E-cigarettes’’ word is adopted to describe the device and the ‘’E-smoking’’ word is used to describe the process of inhalation of the aerosol as described in the first paragraph of the introduction. The e-cigarette word is changed into e-cigarettes throughout the manuscript.

Introduction:

1. Lines 51 – 54: The reference 4 is about prevalence of e-cigarette among high school students, not reference 5!!!!

Response: Thank you for your notice. References are modified accordingly.

2. Lines 54 – 55: Reference 6 is about prevalence among European countries, not globally!!!!

Response: Thank you for your notice. The sentence is modified accordingly.

Methodology:

1. Lines 107 – 108: the sample size calculation is misleading!!!!! It is not indicated whether the response rate and the number of estimates were considered or not, not to mention the “design effects”!!!!! Further, the total population and the population frame were not referred to.

Response: The required details were added to the study design and sampling subsection of the methods.

2. Lines 112 – 113: How can an online questionnaire be signed?????

Response: The questionnaire was constructed using Google Forms. After the link is opened, the first page contains the consent form which describes the study goals and sections. Participants who agreed to participate have to press the agreement icon on the first page to open the questionnaire questions and students who have not agreed to participate, have not completed the entire questions, or have not pressed the submission icon at the end of the questionnaire was not be enrolled in the study. 

3. Line 122 – 123: Merely mentioning that the questionnaire is translated into Arabic with good Cronbach alpha is not enough to use it on an Arabic population. In order to do so, you have to conduct a study on the psychometric properties of the translated version following very strict steps ahead of using it. This is a major methodological pitfall that jeopardizes the whole study.

Response: Thank you for your comment. Psychometric factor analysis was added to the methods section.

Employing a pre-tested questionnaire in Arabic from prior research conducted among university students in Qatar, with clear consent obtained from the author (35). The questionnaire was initially adapted from the Global Adult Tobacco Survey and the American Cancer Society’s Tobacco-Free Generation Campus Initiative: Cohort 5 Student Survey (2020–2021) (36). The Arabic version was used in the current study and the calculated Cronbach’s alpha value for internal consistency was 0.74. Furthermore, the Arabic version of the questionnaire underwent evaluation by five language experts and it was found to be consistent and achieve the target of the study. Additionally, the Kaiser-Meyer-Olkin Measure of Sampling Adequacy indicated that the sample size of the study was appropriate for the factor analysis (0.74), which should be greater than 0.60 (37) to conduct a factor analysis. Bartlett’s Test of Sphericity was also significant (Sig.=0.000<0.05) which means that our variables are related, thus deemed suitable for structure detection and for conducting a factor analysis. The cumulative variance explained by two factors is 53.123% of the total variance as shown from the table below.

Total Variance Explained

Knowledge item Initial Eigenvalues Rotation Sums of Squared Loadings

 Total % of Variance Cumulative % Total % of Variance Cumulative %

1 2.327 33.250 33.250 2.283 32.620 32.620

2 1.391 19.876 53.126 1.435 20.505 53.126

3 0.910 13.001 66.127 

4 0.671 9.592 75.719 

5 0.642 9.171 84.890 

6 0.583 8.325 93.215 

7 0.475 6.785 100.000 

Extraction Method: Principal Component Analysis.

4. Regarding the seven items that gauge the knowledge, I am afraid that there is no sound evidence for the correct responses!!!!!

Response: The correct answers for 4,6, and 7 questions in the knowledge scale were intentionally false. Based on the questionnaire source instruction, every correct answer deserves one point while incorrect answers and ‘’I do not know’’ do not deserve any point. These questions were bolded in Table 2. 

5. What the category “Other” in the “Marital status” mean in an Arabic Culture????

Response: The “Other” in the “Marital status” can be divorced or widowed.

6. Based on what you chose the values of the family income as 516$, 516-1033$ and more than 1033$?????

Response: The abovementioned family income was chosen based on previous studies performed in Palestine such as the following reference: 

‘’Fayyad M, Al-Sinnawi AR. COVID-19 impact on labour relations in Palestine, the need for legal reform. Heliyon. 2021 Nov;7(11):e08313. doi: 10.1016/j.heliyon.2021.e08313. Epub 2021 Nov 4. PMID: 34778590; PMCID: PMC8573059’’

The current used currency is converted to dollars in our paper to make it easier for readers.

Results:

1. Indicate that the responses to “Reasons for using e-cigarettes” are multiple choices.

Response: The suggestion is indicated at the end of the study tools section.

2. Table 5: Correct the percentage 991.1%!!!!

Response: Done

3. Why you included the knowledge as a determinant in the logistic regression??? having a lower knowledge might be a consequence of being e-cigarette users, either as a direct association, or indirectly through ignoring and indifference to its bad effects.

Response: knowledge score and other significant variables in the bivariate analysis were included in the logistic regression analysis to evaluate whether having low knowledge is associated with being an e-cigarette user.

Discussion:

1. The discussion is superficial. For example, the authors didn’t comment on why there were differences in knowledge between e-cigarette users and non-users!!!!!!!

Response: The discussion was modified and further details were included.

2. Many of the arguments are not supported by references!!!!!

Response: Modifications with proper citations are added.

3. Protrude your study’s strengths, and mention its limitations.

Response: Done

4. The conclusion is not relevant to the study at all.

Response: The conclusion is modified accordingly.

---

## [Decision Letter · Decision Letter 1]

16 Apr 2024

E-cigarette use among University Students in Palestine: Prevalence, Knowledge, and Determinant Factors

PONE-D-24-02812R1

Dear Dr. Ghanim,

We’re pleased to inform you that your manuscript has been judged scientifically suitable for publication and will be formally accepted for publication once it meets all outstanding technical requirements.

Kind regards,

Mohammed Nasser Alhajj, BDS, MClinDent, PhD

Academic Editor

PLOS ONE

Additional Editor Comments (optional):

Reviewers' comments:

Reviewer's Responses to Questions

**Comments to the Author**

1. If the authors have adequately addressed your comments raised in a previous round of review and you feel that this manuscript is now acceptable for publication, you may indicate that here to bypass the “Comments to the Author” section, enter your conflict of interest statement in the “Confidential to Editor” section, and submit your "Accept" recommendation.

Reviewer #1: All comments have been addressed

Reviewer #3: All comments have been addressed

2. Is the manuscript technically sound, and do the data support the conclusions?

Reviewer #1: Yes

Reviewer #3: Yes

3. Has the statistical analysis been performed appropriately and rigorously? 

Reviewer #1: Yes

Reviewer #3: Yes

4. Have the authors made all data underlying the findings in their manuscript fully available?

Reviewer #1: No

Reviewer #3: Yes

5. Is the manuscript presented in an intelligible fashion and written in standard English?

Reviewer #1: Yes

Reviewer #3: Yes

6. Review Comments to the Author

Reviewer #1: No more comments.

All the comments have been addressed by the authors.

I would recommend accepting the manuscript for publication in the journal .

Reviewer #3: (No Response)

7. PLOS authors have the option to publish the peer review history of their article (what does this mean?). If published, this will include your full peer review and any attached files.

Reviewer #1: No

Reviewer #3: **Yes: **Esam Halboub
